# Cytomolecular Classification of Thyroid Nodules Using Fine-Needle Washes Aspiration Biopsies

**DOI:** 10.3390/ijms23084156

**Published:** 2022-04-09

**Authors:** Giulia Capitoli, Isabella Piga, Vincenzo L’Imperio, Francesca Clerici, Davide Leni, Mattia Garancini, Gabriele Casati, Stefania Galimberti, Fulvio Magni, Fabio Pagni

**Affiliations:** 1Bicocca Bioinformatics Biostatistics and Bioimaging B4 Center, Department of Medicine and Surgery, University of Milano-Bicocca, 20900 Monza, Italy; giulia.capitoli@unimib.it (G.C.); stefania.galimberti@unimib.it (S.G.); 2Proteomics and Metabolomics Unit, Department of Medicine and Surgery, University of Milano-Bicocca, 20900 Monza, Italy; isabella.piga@unimib.it (I.P.); francesca.clerici@unimib.it (F.C.); fulvio.magni@unimib.it (F.M.); 3Department of Medicine and Surgery, Pathology, University of Milano-Bicocca, 20900 Monza, Italy; vincenzo.limperio@gmail.com; 4Department of Radiology, San Gerardo Hospital, ASST Monza, 20900 Monza, Italy; daleni@tin.it; 5Department of Surgery, San Gerardo Hospital, ASST Monza, 20900 Monza, Italy; mattia_garancini@yahoo.it; 6Department of Clinical Pathology, San Gerardo Hospital, ASST Monza, 20900 Monza, Italy; gabriele.casati94@gmail.com

**Keywords:** MALDI-MSI, thyroid carcinoma, proteomic analysis, diagnostic classification, fine-needle aspiration biopsies

## Abstract

Fine-needle aspiration biopsies (FNA) represent the gold standard to exclude the malignant nature of thyroid nodules. After cytomorphology, 20–30% of cases are deemed “indeterminate for malignancy” and undergo surgery. However, after thyroidectomy, 70–80% of these nodules are benign. The identification of tools for improving FNA’s diagnostic performances is explored by matrix-assisted laser-desorption ionization mass spectrometry imaging (MALDI-MSI). A clinical study was conducted in order to build a classification model for the characterization of thyroid nodules on a large cohort of 240 samples, showing that MALDI-MSI can be effective in separating areas with benign/malignant cells. The model had optimal performances in the internal validation set (*n* = 70), with 100.0% (95% CI = 83.2–100.0%) sensitivity and 96.0% (95% CI = 86.3–99.5%) specificity. The external validation (*n* = 170) showed a specificity of 82.9% (95% CI = 74.3–89.5%) and a sensitivity of 43.1% (95% CI = 30.9–56.0%). The performance of the model was hampered in the presence of poor and/or noisy spectra. Consequently, restricting the evaluation to the subset of FNAs with adequate cellularity, sensitivity improved up to 76.5% (95% CI = 58.8–89.3). Results also suggest the putative role of MALDI-MSI in routine clinical triage, with a three levels diagnostic classification that accounts for an indeterminate gray zone of nodules requiring a strict follow-up.

## 1. Introduction

The cytological evaluation of fine-needle aspiration (FNA) biopsy is currently the gold standard to exclude the malignant nature of thyroid nodules [1]. However, approximately 20–30% of FNAs have an “indeterminate for malignancy” final report, representing an important problem in terms of pursuing standardized treatments and preventing unnecessary thyroid surgery [2]. A very significant number of patients with thyroid nodules after biopsy undergo diagnostic and not therapeutic thyroidectomy, with important implications in terms of healthcare costs, operative risks and morbidity, and the need for a lifelong hormone replacement therapy [3]. Therefore, the identification of powerful diagnostic tools for assisting cytopathologists in the appropriate diagnosis of thyroid lesions is a clinical, ethical, and economically relevant issue [4].

Different molecular tests pointing to abnormal molecular mechanisms of thyroid cancer, such as genetic testing (BRAF, N-H-KRAS point mutations, and RET/PTC1, RET/PTC3, PAX8/PPAR rearrangements) or gene-expression classifiers (Veracyte, San Francisco, CA, USA, Thyroseq, Rye Brook, NY, USA), have been proposed to improve the preoperative risk assessment of malignancy on thyroid FNAs [5], but their performances are variable based on the markers tested (rule-in vs. rule-out tests) and their costs still limit the implementation in the routine clinical practice.

Matrix-assisted laser desorption ionization mass spectrometry imaging (MALDI-MSI) is a label-free, non-destructive technology that explores the spatial distribution of biomolecules both on tissue sections and cytological samples [6,7], combining the analytical power of mass spectrometry with traditional light microscopy. In the last ten years, MALDI-MSI has been one of the key techniques for cancer biomarker discovery directly in-situ by using statistical and machine learning approaches [8,9]. The application of MALDI-MSI to thyroid FNA is an important challenge with intriguing clinical potentials and has been recently approached with relevant results in terms of feasibility and technical reproducibility. A robust MALDI-MSI sample preparation workflow evaluating the stability of thyroid FNAs over time was optimized, and the hemoglobin interference of bloody FNAs was successfully overcome [10,11].

Here, we describe the application of MALDI-MSI proteomic analysis to thyroid needle washes FNAs to assist cytopathologists in the diagnostication of indeterminate cases with the final aim of avoiding unnecessary surgery.

## 2. Results

### 2.1. Patient and Nodule Characteristics

A total of 207 patients contributed with 240 FNAs: 220 needle washes from thyroid nodules (17 from a multinodular context and 3 from lymph nodes) and 20 dedicated FNAs. The majority of patients were female (74.4%) and Caucasian (97.1%); the median age was 60 years (Q1–Q3 = 50.0–70.5). Subjects were mostly with no therapy at FNA (82%), and there was a low rate of familial history of thyroid cancer (5.8%). Thyroid biopsy was done in 106 (51.2%) single and 93 (44.9%) multinodular contexts, and the mean nodule size was 20.7 mm. Details on nodule characteristics are reported in Appendix A.

### 2.2. MALDI-MSI Spectra Characteristics

The results of the PCA on the 240 MALDI-MSI overall average spectra suggested that the degree of clustering in the benign nodules (i.e., Hashimoto thyroiditis (HT), hyperplastic (HP), and HP-associated lymph node) is high, as compared to the heterogeneity observed in the malignant group (i.e., medullary thyroid carcinoma (MTC), papillary thyroid carcinoma (PTC), and PTC-associated lymph node) (Appendix A). Borderline cases (i.e., non-invasive follicular thyroid neoplasm with papillary-like nuclear features (NIFTP), follicular adenoma (FA), and well-differentiated tumor of uncertain malignant potential (WDT-UMP)) shared spectra profiles with both the benign and malignant groups.

### 2.3. MALDI-MSI to Classify Thyroid Nodules: Training Set

We first built a predictive model to discriminate PTC from HP and HT using 605 ROIs (381 HP, 50 HT, 174 PTC) extracted from 70 FNAs. Among the 95 *m*/*z* features selected by the Lasso model (Appendix A), we reported the MALDI-MSI localization on cytology of the six *m*/*z* that mostly contributed to classification (Appendix A). The internal validation of the model had an accuracy of 94.4% and led to the correct classification of 96.1% HP (366/381), 76.0% HT (38/50, the remaining 12 were assigned to HP), and 96.0% PTC ROIs (167/174), respectively (Appendix A). Remarkably, 97.2% of the benign ROIs (419/431) were correctly classified when HT and HP were considered as a unique benign entity. Moreover, the results of the diagnostic performances on the per-nodule classification using either pixel-by-pixel (on 860.637 pixels, and with 16.7% of the pixels as a cut-off for malignancy), ROIs, or overall average spectra are reported in Appendix A. The approach using ROIs fully agreed with histopathology/follow-up, while the remaining two had sensitivity ranging from 95.0% to 100% and specificity from 96.0% to 100% (Appendix A). In the ROIs approach, the presence of at least one-third of ROIs classified as malignant, within sample, was used as a threshold to classify the nodule as malignant. A graphical representation of the pixel-by-pixel results of the classifier is reported in Figure 1a. Of note, the mean rate of pixels misclassified in the 50 TIR2/THY2 nodules (630,383 pixels) was 2.0% (SD = 5.2%), while in the 20 PTC samples (230,254 pixels), the mean rate of pixels correctly classified as malignant was 89.7% (SD = 19.5) (Figure 1b).

### 2.4. MALDI-MSI to Predict Diagnosis: Validation Set

We assessed the performances of the classifier identified by the Lasso model on an independent set of 170 FNAs (corresponding to 1815 ROIs and 2,178,297 pixels). The results of the per-nodule analysis based on ROIs and pixel-by-pixel were similar (Table 1), with a common specificity of 82.9% (95% CI = 74.3–89.5%) and a sensitivity of 40.0% (95% CI = 28.0–52.9%) and 43.1% (95% CI = 30.9–56.0%), respectively.

Out of 65 malignant samples in the pixel-by-pixel analysis, 37 false negatives were PTC, of which 29 revealed non-satisfactory cellular contents; on the contrary, 26 out of the 28 true positive lesions had adequate cellularity. Conversely, among the benign samples, only 4 out of 18 false positives presented inadequate sample quality, and the main errors on adequate samples (*n* = 14) can be partly attributed to the presence of seven borderline entities, such as one NIFTP and six FA. Of the remaining seven benign cases, two were HT, three HP, one Goiter, and the last one was taken from a peri-thyroid lymph node and therefore entirely composed of lymphocytes without benign thyrocytes. Focusing on indeterminate cases, 80.9% (55/68) of TIR3/THY3 with a final benign histology/follow-up were correctly classified, while all nine TIR3/THY3 with a malignant histological diagnosis were misclassified (Appendix A). Moreover, of 19 TIR4/THY4, 66.7% (10/15) were correctly indicated as malignant and 75% (3/4) correctly as benign, as per final histology/follow-up (Appendix A).

Removing cases with “borderline” histology (i.e., NIFTP, FA, WDT-UMP) from the validation cohort, specificity increased to 88.2% (95% CI = 78.7–94.4%), with unchanged sensitivity for the pixel-by-pixel analysis on 141 samples and with similar results using the ROIs approach (Table 1). Further restricting the analysis to a subset of 72 cases with adequate cellularity, the sensitivity increased to 76.5% (95% CI = 58.9–89.3%) with a specificity of 81.6% (95% CI = 65.7–92.3%) on the pixel-by-pixel approach. Specifically, the model classified 57 nodules correctly with respect to histology/follow-up, with a correct assignment in the 82.4% (14/17) of the benign TIR2/THY2 and 73.9% (17/23) of the malignant TIR5/THY5. Remarkably, the model successfully classified 17 out of 21 benign pre-operative indeterminate nodules, yielding a specificity and NPV of 81.0% (95% CI = 58.1–94.6%) and 89.5% (95% CI = 66.9–98.7%), respectively. In addition, the diagnoses of the nine TIR4/THY4 cases, which were all malignant at histology, were confirmed at model prediction.

### 2.5. Novel Diagnostic Workflow Based on MALDI-MSI Analysis

We propose a novel workflow that combines cytopathology and MALDI-MSI analysis in the diagnosis of thyroid nodules for implementation in clinical practice. This relies on the performances of the pixel-by-pixel approach that have been investigated using a three-level diagnostic classification strategy based on selected percentages of malignant pixels (namely <7.0%, 7.0–16.7%, and >16.7%). The results of this analysis are reported in Appendix A. In this setting, 78.8% of TIR2/THY2 (26/33) were correctly identified, and this rate raised to 87.9% (29/33) considering the three cases in the “gray-zone” (7–16.7%) as benign after adequate follow-up. Only 4 out of 33 TIR2/THY2 were misclassified: three HT and a peri-thyroid lymph node that did not contain benign thyrocytes, being entirely composed of lymphocytes. The detection rate of benign cases in the TIR3/THY3 group was 76.5% (52/68), reaching 80.9% (55/68) considering “gray-zone” cases (Appendix A). The 66.7% (10/15) and 43.9% (18/41) of the agreement for malignancy characterization were reached in TIR4/THY4 and TIR5/THY5 patients. Moreover, focusing on the 45 nodules with indeterminate cytology and adequate cellularity, we observed 90.9% of agreement in malignant diagnosis (considering the 1 case in the “gray-zone”) and 81.0% (17/21 with HP diagnosis) in benign nodules (Table 2).

## 3. Discussion

Previous experiences already investigated with favorable achievements the capability of mass spectrometry in detecting malignant thyroid nodules on pathological tissues [12,13,14], FNAs smear [12,15], or dedicated FNA samples collected after thyroidectomy [7]. The natural evolution of these research studies has been the adoption of FNA needle washes. Preliminary results on this source were encouraging [16,17,18,19], showing that specific molecular profiles characterize different types of thyroid lesions. Our large prospective study (240 lesions/FNAs) had the aim to develop and validate a classification model using mass spectrometry data collected via MALDI-MSI to discriminate PTC from benign nodules, improving the diagnosis of indeterminate cases.

The internal validation of our classification model showed very good performances (accuracy = 97.1%, AUC = 0.998, sensitivity = 100.0%, and specificity = 96%). In an independent validation set, high specificity (82.9%, 95% CI = 74.3–89.5) but low sensitivity (43.1%, 95% CI = 30.9–56.0) were achieved with the pixel-by-pixel approach that excludes the intervention of the pathologist in annotating ROIs. Many of the false-positive cases with adequate cellularity (10/14) had a clinical diagnosis of HT, FA, NIFTP or were lymph nodes, and the model might have recognized areas with atypical thyrocytes or with incipient tumor progression or did not recognize the “usual” thyroid cells [17,19]. Conversely, false-negative results depended on FNA quality since the majority (29/37) had poor cellularity on ITO slides as assessed post MALDI analysis. Remarkably, the predictive model was optimal in detecting cases with final histology different from papillary carcinomas such as MTC and rare parathyroid tumors (7/7). Several are the explanations for false negatives: (1) the relative scarcity of diagnostic cells in the specimens can significantly impact the final results of MALDI-MSI analysis [16]; (2) the relative proportion of malignant cells in an otherwise benign background can reduce the percentage of malignant pixels affecting the final classification of the model, as exemplified by two cases of macrophages-flood cystic PTC [18]. Moreover, the distribution of malignant cells in the sample can be heterogeneous, presenting sparse and/or overlapping cellular content. FNA spots composed of a few clusters of carcinomatous thyrocytes in regions with dominant different histologic features can be insufficient for the model to define the whole sample as malignant, as demonstrated by the variability reported in the PCA analysis (Appendix A). Cytological evaluation of the ITO slide after MALDI-MSI analysis confirmed the impact of cellular quality on the discriminant performance of malignant samples. Indeed, in the subset of cases with adequate cellularity, sensitivity increased to 76.5% (95% CI = 58.9–89.3) and PPV to 78.8% (95% CI = 61.1–91.0) (Table 1). In particular, when focusing on the 23 indeterminate cytological diagnoses, a prediction of non-malignancy was obtained in 17 out of 21 cases that agreed with the 2-year follow-up or the final histopathological diagnosis. In addition, nine patients with a suspicious diagnosis resulted in being malignant in agreement with post-surgical pathological evaluation. Our results on FNA needle washes that contain adequate thyroid cells or MALDI-MSI signals are not completely in line with those obtained by DeHoog, 2019 [12], who achieved a sensitivity of 96% (95% CI = 79–99%) and a specificity of 91% (95% CI = 77–97%) on 58 FNA smears, focusing on lipid content, using a pixel-by-pixel approach. It is important to highlight that (1) DeHoog used a molecular approach based on metabolic analysis (desorption electrospray ionization mass spectrometry, DESI-MS) on FNA smears; and (2) when considering the precision in the estimates, DeHoog’s and our 95% CIs overlap. Overall, the results obtained in the external validation set showed a “rule-out” role for MALDI-MSI analysis, suggesting that it might be useful in the routine clinical triage of thyroid nodules, appropriately detecting benign cases, especially in the setting of indeterminate cytology. This approach shows similarities with the one recently proposed for new genetic tests with high specificity and NPVs [20], in which complementary analyses are indicated to support pathologists, who are still central in the diagnostic process since they are highly reliable in distinguishing clear-cut benign/malignant cases. In addition, we suggest a three levels diagnostic classification for the “indeterminate” nodules in the pixel-by-pixel approach that excludes (<7.0% of malignant pixels) or confirms (>16.7%) malignancy with high probability, thus accounting for an intermediate gray zone (7.0–16.7%) that identifies nodules requiring a strict US follow-up, eventually associated with a repeat biopsy. This workflow is potentially transferable in clinical practice (Figure 2) since the cytological diagnosis is rather rapid and needle washes from FNA samples can be preserved for up to two weeks, as previously shown in Piga et al.’s work [10], making sure that, when the sample is not adequate for MALDI-MSI, a repeat FNA is required. Importantly, the implementation of MALDI-MSI would also have a positive effect on healthcare costs with respect to diagnostic, but not therapeutic thyroidectomy, that has a great impact both on psychological health and the potential need for a lifelong hormone replacement drug therapy for patients. Indeed, once the MALDI-MSI approach becomes a first-line diagnostic tool for indeterminate cases in the routine workflow of thyroid FNA, an easy-to-use system will be realized, similarly to what has been already implemented in microbiology units (MALDI-Biotyper®, Bruker Daltonics, Bremen, Germany [21]). The system will thus increase diagnostic accuracy while reducing the costs related to unnecessary surgery and will improve the cost-effectiveness of post-treatment follow-up examinations.

We are aware of the challenging aspects of the present work. First, the predictive values we obtained are largely dependent on the prevalence of malignancy at our institution; thus, further validations are needed on different cohorts to better clarify the performances of MALDI-MSI. Moreover, although we used a large number of ROIs to train the model, the narrow spectrum of malignant cases might have potentially affected the capabilities of our classification model to recognize carcinomas signatures.

## 4. Materials and Methods

### 4.1. Patients

A cohort of 207 consecutive patients admitted to the ultrasound (US)-guided FNA clinic of the ASST Monza (San Gerardo Hospital, Università di Milano-Bicocca, Monza, Italy) was enrolled from 1 January 2017 to 30 June 2019 in a prospective trial for marker discovery in thyroid cancer. Patients underwent a standard procedure of US-guided FNA that included a minimum of 2 needle passes per nodule. Both Papanicolaou and Hematoxylin & Eosin (H&E) staining were prepared. A morphological FNA diagnosis according to the 5-tiered SIAPEC/Bethesda reporting systems [2,22] was obtained by 2 expert cytopathologists. After FNA, patients with a non-malignant diagnosis (SIAPEC/Bethesda = TIR2/THY2, TIR3A/THY3A) underwent a US examination every 12-months to exclude the presence of echographic malignant features, i.e., absence of (a) significant increase in nodule size (>15%), (b) nodes metastasis, and (c) new suspicious nodules [23,24]. A minimum follow-up of 24 months after enrollment was used to confirm the benign nature of nodules. Instead, patients with a malignant or indeterminate cytological diagnosis (SIAPEC/Bethesda > TIR3A/THY3A) underwent thyroidectomy and were classified according to the latest World Health Organization classification of endocrine tumors based on the final histology after surgery [25]. Due to the introduction of the new non-invasive follicular thyroid neoplasm with papillary-like nuclear features (NIFTP) category, while the trial was ongoing [26], the encapsulated follicular variant of papillary thyroid carcinoma (PTC) diagnoses was re-evaluated and reclassified appropriately.

The study was approved by the Ethical Board of the ASST Monza (AIRC MFAG 2016, n.133/7-2-2017). All patients signed informed consent.

### 4.2. Training and Validation Set

Seventy samples with adequate cellularity (i.e., good-optimal) were used in the training phase to minimize the risk of potential interferences in model building (e.g., poor cellularity, low tumoral/malignant component in an otherwise benign background). Sample cellularity was assessed semiquantitatively by one cytopathologist as follows: poor (when the sample contained 20–30% of thyrocytes cells), good (with 30–70%), and optimal (>70%). We included in the training set 50 TIR2/THY2 benign nodules (corresponding to 40 hyperplastic -HP- and 10 Hashimoto thyroiditis -HT) and 20 TIR5/THY5 malignant papillary thyroid carcinomas (PTCs). The validation set consisted of 170 cases: 33 TIR2/THY2, 77 TIR3/THY3, 19 TIR4/THY4, and 41 TIR5/THY5. Diagnosis from histology or follow-up according to cytological classes is reported in Appendix A.

### 4.3. MALDI-MSI

Both types of samples, needle washes (left-over material in the syringe washed out in the falcon) and dedicated passes of the FNA, were collected into a CytoLyt solution (20% buffered methanol-based solution, ThinPrep® CytoLyt system, CYTYC Corporation, Hologic, Marlborough, MA, USA) and prepared as previously described [10]. Then, they were transferred on Indium Tin Oxide (ITO) slides directly. Slides have been vacuum desiccated and sent to the MALDI-MSI examination. Each slide was washed, and sinapinic acid was used as a matrix for MALDI-MSI analysis. All the mass spectra were acquired in linear positive mode in the mass range of 3000–15,000 *m*/*z*, using 300 laser shots per spot, with a laser focus setting of 50 μm and a pixel size of 50 × 50 μm with an UltrafleXtreme mass spectrometer (Bruker Daltonics, Bremen, Germany). A mixture of standard proteins within the mass range of 5730–16,950 Da (Protein Calibration Standard I. Bruker Daltonics, Billerica, MA, USA) was used for external calibration (mass accuracy ± 30 ppm). Data acquisition and visualization were performed using the Bruker software packages (flexControl v3.4, flexImaging v5.0, Bruker Daltonics, Bremen, Germany). After MALDI-MSI analysis, the matrix was removed as previously described and the slide was stained with H&E. Finally, the cytological specimen was converted to digital by scanning the slide through the ScanScope CS digital scanner (Aperio, Park Center Dr., Vista, CA, USA), thus allowing the direct overlap and integration of cytomorphological and molecular information. Regions of interest (ROIs) containing pathological areas of interest (i.e., benign and malignant thyrocytes, inflammatory background) were comprehensively annotated by the pathologist.

### 4.4. Statistical Analysis

Continuous characteristics were described as mean and standard deviation (SD) or quartiles (Q1, median, and Q3), as appropriate, while qualitative variables were reported as count and frequency.

After spectra preprocessing (see Appendix B for details), an exploratory analysis of nodules’ average spectra was performed to assess similarities through a principal component analysis (PCA) on scaled and centered data. We applied a multinomial penalized regression with the Lasso regularization model on the ROIs average spectra of the training set to pick the *m*/*z* signals able to distinguish malignant cells (PTC) from normal thyrocytes (HP) or inflammatory background (HT). Cross-validation (5-fold) was performed to select the Lasso penalizing parameter and to assess predictive accuracy. The resulting classification model was used to predict the probability to belong to one of the three classes above using the mass spectra from the validation set. Validation was performed using three different approaches: (1) overall average spectrum, (2) ROI average spectrum, and (3) pixel-by-pixel (single spectra analysis). Specifically, the per-nodule diagnosis (benign vs. malignant) was defined by: (1) the highest of the three probabilities (i.e., HP, HT, and PTC) on the overall average spectrum, (2) the presence of at least one-third of ROIs classified as malignant, and (3) the percentage value of malignant pixels above the threshold obtained by the Youden Index derived on the pixel-by-pixel analysis in the training set.

A three levels diagnostic classification, also including a gray diagnostic zone on top of the usual dichotomic classification in benign vs. malignant, was considered, and a double cut-off was estimated, maximizing sensitivity and specificity on the training set.

The validation phase was performed blinded to the patient’s histological diagnosis. Sensitivity, specificity, positive predictive value (PPV), negative predictive value (NPV), and accuracy were calculated along with the corresponding 95% confidence intervals (CI).

Data preprocessing (MALDIquant package) and statistical analyses were performed using the open-source R software v.3.6.0 (R Foundation for Statistical Computing, Vienna, Austria).

## 5. Conclusions

Our results provided evidence that MALDI-MS imaging on needle washes FNAs is a promising approach to aid in the diagnosis of thyroid nodules, on the condition that the cytological sample is cellularly adequate. Further research is needed to strengthen our classification model, but these findings are suggestive that it might be worth introducing MALDI-MS imaging into the clinical diagnostic workflow of thyroid neoplasia. This would boost clinical management, especially in cases of indeterminate cytology limiting diagnostic surgeries and, ultimately, the outcomes of patients.

## Figures and Tables

**Figure 1 ijms-23-04156-f001:**
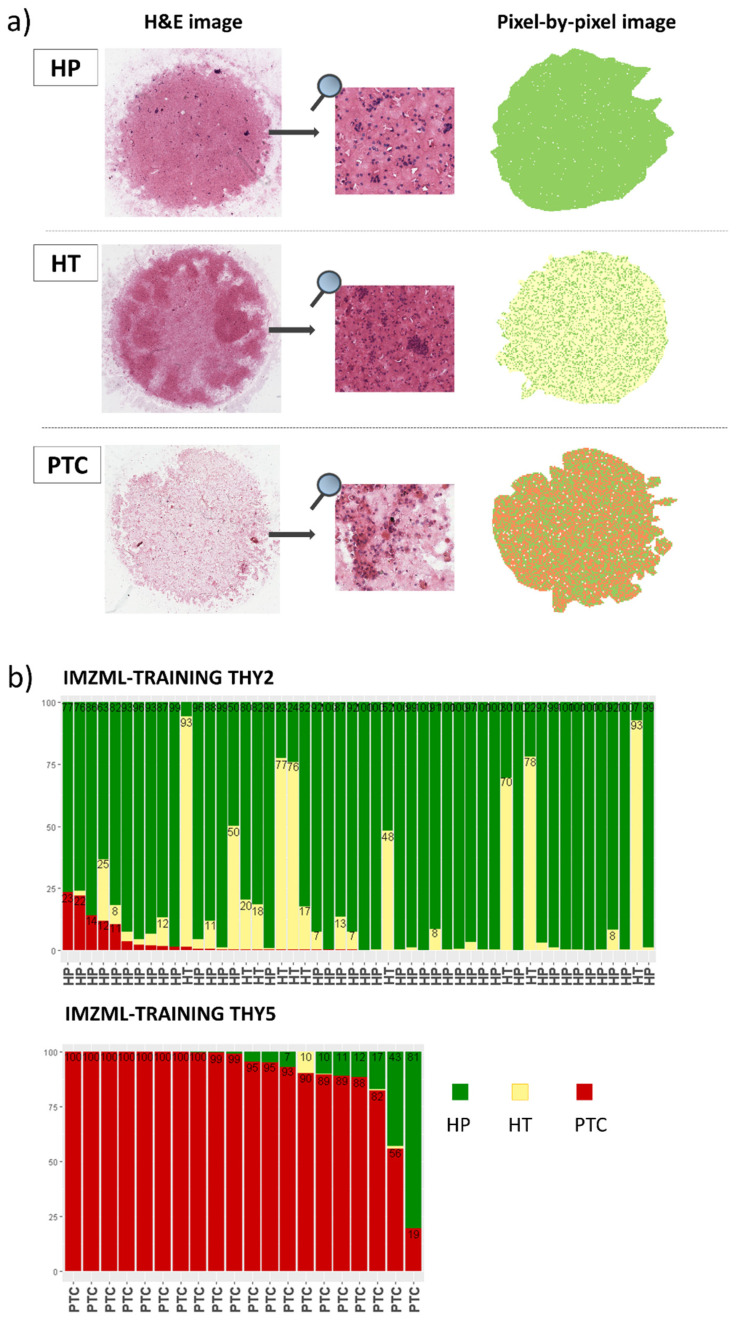
Results of the pixel-by-pixel (IMZML) classification of the training set samples. (**a**) Examples of H&E and pixel-by-pixel images of hyperplastic (HP), Hashimoto’s thyroiditis (HT), and papillary thyroid cancer (PTC) samples; (**b**) stacked bar charts of the percentage of pixels in each FNA of the training set classified as HP (green), HT (yellow), and PTC (red) in the 50 TIR2/THY2 and 20 TIR5/THY5 samples. Each FNA was classified as malignant when the percentage of red pixels was >7%.

**Figure 2 ijms-23-04156-f002:**
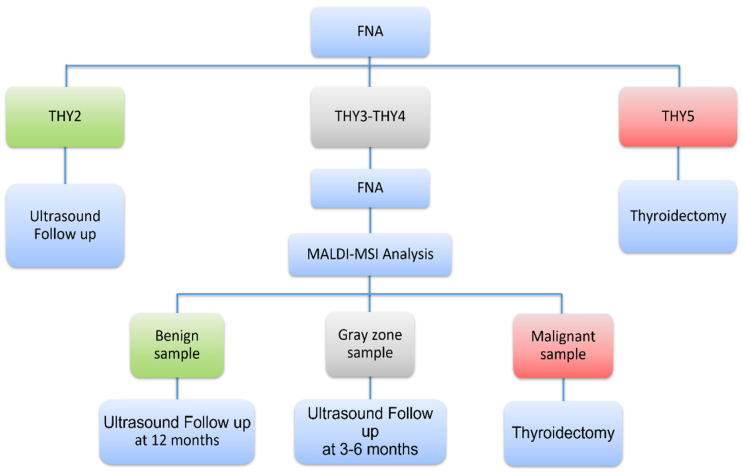
Diagnostic flow-chart combining cytopathology and MALDI-MSI proteomic analysis.

**Table 1 ijms-23-04156-t001:** Performances of the classifier on the three MALDI-MSI approaches in the validation set, overall and by subgroups.

	TP	FP	TN	FN	% Sensitivity (95% CI)	% Specificity (95% CI)	% PPV (95% CI)	% NPV (95% CI)	% Accuracy
**Overall (*n* = 170)**									
**Pixel-by-pixel**	28	18	87	37	43.1 (30.9–56.0)	82.9 (74.3–89.5)	60.9 (45.4–74.9)	70.2 (61.3–78.0)	67.7
**ROIs**	26	18	87	39	40.0 (28.0–52.9)	82.9 (74.3–89.5)	59.1 (43.3–73.7)	69.1 (43.3–73.7)	66.5
**Average spectra**	20	15	90	45	30.8 (19.9–43.5)	85.7 (77.5–91.8)	57.1 (39.4–73.7)	66.7 (58.0–74.5)	64.7
**Borderline diagnosis ° excluded (*n* = 141)**									
**Pixel-by-pixel**	28	9	67	37	43.1 (30.9–56.0)	88.2 (78.7–94.4)	75.7 (58.8–88.2)	64.4 (54.4–73.6)	67.4
**ROIs**	26	10	66	39	40.0 (28.0–52.9)	86.8 (77.1–93.5)	72.2 (54.8–85.8)	62.9 (52.9–72.1)	65.3
**Average spectra**	20	7	69	45	30.8 (19.9–43.5)	90.8 (81.9–96.2)	74.1 (53.7–88.9)	60.5 (50.9–69.6)	63.1
**Samples with adequate cellularity (*n* = 72) ***									
**Pixel-by-pixel**	26	7	31	8	76.5 (58.8–89.3)	81.6 (65.7–92.3)	78.8 (61.1–91.0)	79.5 (63.5–90.7)	79.2
**ROIs**	21	8	30	13	61.8 (43.6–77.8)	79.0 (62.7–90.5)	72.4 (52.8–87.3)	69.8 (53.9–82.8)	70.8
**Average spectra**	19	5	33	15	55.9 (37.9–72.8)	86.8 (71.9–95.6)	79.2 (57.9–92.9)	68.8 (53.8–81.3)	72.2

* 17 TIR2/THY2, 23 TIR3/THY3, 9 TIR4/THY4, and 23 TIR5/THY5; ° NIFTP, adenoma and WDT-UMP. Legend: TP—true positive; FP—false positive; TN—true negative; FN—false negative; PPV—positive predictive value; NPV—negative predictive value; 95% CI—95% confidence interval.

**Table 2 ijms-23-04156-t002:** Predicted diagnosis based on the three-level diagnostic classification strategy in the pixel-by-pixel analysis of the 45 nodules with indeterminate cytology and adequate cellularity according to histopathology/follow-up.

Histopathology/Follow-Up	Total	Predicted Diagnosis		
		Benign	Gray Zone	Malignant
**Benign (*n* = 34)**				
**HP**	21	17 *	0	4 °
**HT**	0	0	0	0
**FA**	9	3	0	6
**NIFTP**	3	2	0	1
**WDT-UMP**	1	1	0	0
**Malignant (*n* = 11)**				
**PTC**	8	1	1	6 ^#^
**MTC**	3	0	0	3

* 4 Goiter; ° 1 Goiter; ^#^ 2 rare parathyroid tumor. HP—hyperplastic; HT—Hashimoto thyroiditis; FA—follicular adenoma; PTC—papillary thyroid carcinoma; MTC—medullary thyroid carcinoma; WDT-UMP—well-differentiated tumor of uncertain malignant potential; NIFTP—non-invasive follicular thyroid neoplasm with papillary-like nuclear features.

## Data Availability

Data that support the findings of this study are available on request from the corresponding author F.P. upon reasonable request.

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
