# Peer review of "Cytomolecular Classification of Thyroid Nodules Using Fine-Needle Washes Aspiration Biopsies"

_ijms, 2022, doi:10.3390/ijms23084156_

Round 1
Reviewer 1 Report
Introduction
- Please do reffer to the current accepted/well documented aplucationf of MALDI, in different neoplasical diseases such as prostatic cancer, pituitary neoplasia,.
RESULTS
2.1 please define clearly what do you mean by 220 needle washes and 20 dedicated FNAs.
I would suggest tot o mix the results from the 3 lypmh nodes in the final analysis.
Do desribe in the multinodular goiter show did you choos which nodule to referr to FNA.
Why did you mention the mean thyroid nodule size?
2.2 Please describe what HT, HP, PTC, MTC means.
2.3 More clarification about defining the ROI.s (605) from 70 FNA=s are needed. Did you use a fix number of ROI for each FNA sample? Din you use a ROI for each FNA atempt or from 1 FNA sample per patientyou defined different ROIs? It is really unclear the selection modality used for the predictive model of discrimination.
Please desribe the selection of ROIs in pixel by pizel approach.
Did you use PATHOLOGY report for the final analysis in order to define benign versus malignant tissue? Did you exclude from the analysis cases with microcarcinoma, in which the FNA sample could have missed the cancer spot, and to influnec the final accuracy of the method ( to analyse a bening lession considered due to the adiacent microcarcinoma a final cancer case?)
What do you understand under ........ fulle agrees with histopathology/follow up... ( page 2 R 96)
2.3 Training set =
You are stating that 70 FNAs , from 70 different nodules were used for the analysis. If the different ROI.s used to analyse each nodule, offered diveregent results, how was the final decision = majority, worse scenario, minimal thershold. This detail is really under in this section.
2.4
What is the reason for removing the borderline lessions from the final analysis? In the diagnostic, the use of MALDI MSI is usefull in pre preoperative stage, when there is no clue regarding the final result. The validation of the method should be made on all casses, since AF, NIFTP and WDT UMP can be seen under BETHESDA III and IV citological results. Excluding thees ecase in the validation does falsy increase the accuracy of the method.
- Discussions
In intermediate cytology cases,2 years of follow with is not definitivelly a certitude for benign lessions. When you try to validate a new method/technoque, referral must be made with definitive diagnostic cases.
A compoarision bewteen pixel by pixel approach with the evaluation of lipid content ( as the DeHoog research).
I do support that the technique could find an application in reconfirmim benign lessions, even if the diagnostic capacity of PTC versus AF/NIFT P is difficult.
Author Response
Response to Reviewer 1
INTRODUCTION
Point 1: Please do reffer to the current accepted/well documented aplucationf of MALDI, in different neoplasical diseases such as prostatic cancer, pituitary neoplasia,.
Response 1: Thank you for the suggestion. We modified the introduction accordingly, and two references on prostate cancer and pituitary adenomas has been included:
“Matrix assisted laser desorption ionization mass spectrometry imaging (MALDI-MSI) is a label-free, non-destructive technology that explores the spatial distribution of biomolecules both on tissue sections and cytological samples (6, 7), combining the analytical power of mass spectrometry with traditional light microscopy. In the last ten years, MALDI-MSI has been one of the key techniques for cancer biomarker discovery directly in-situ by using statistical and machine learning approaches (8, 9).”
RESULTS
Point 2.1: Please define clearly what do you mean by 220 needle washes and 20 dedicated FNAs.
I would suggest tot o mix the results from the 3 lypmh nodes in the final analysis.
Do desribe in the multinodular goiter show did you choos which nodule to referr to FNA.
Why did you mention the mean thyroid nodule size?
Response 2.1: Based on your suggestions:
- We clarified the meaning of needle washes and dedicated FNAs in the Materials and Methods section (see section4.3 on MALDI-MSI, page 8, lines 290-291).
- For the sake of completeness, we decided to include in the primary analysis ALL the clinical entities represented in our study, including lymph nodes. When we made the sensitivity analysis, we discarded samples without adequate cellularity and borderline diagnosis, deciding to maintain the three lymph nodes because they have adequate cellularity and their diagnosis was clear-cut. However, we have done the supplemental analysis you suggest, excluding also the three lymph nodes, and the results have not changed.
- Actually we performed FNA in the context of multinodular goiter using as criteria the prevalent nodule size, the ultrasound features of the nodule (echogenicity, calcification, margins, shape and vascularity) and the dimensional changes over time, as per our recent experience (https://pubmed.ncbi.nlm.nih.gov/34771602/)
- We reported the mean nodule size to stress the field of action of the proposed MALDI-based approach. Indeed, in this first prospective report we tried to maximally reduce the rate of inadequate/THY1 cases by increasing the size of nodules, which can make it less prone to inadequate results of the cytology. Thus, at least in this preliminary report, the approach we proposed has its main application on nodules of around 2 cm diameter on average.
Point 2.2: Please describe what HT, HP, PTC, MTC means.
Response 2.2: We thank the reviewer for highlighting this omission. The acronyms are now explicit along the text (see page 2, line 78-83).
Point 2.3: More clarification about defining the ROI.s (605) from 70 FNA=s are needed. Did you use a fix number of ROI for each FNA sample? Din you use a ROI for each FNA atempt or from 1 FNA sample per patientyou defined different ROIs? It is really unclear the selection modality used for the predictive model of discrimination.
Please desribe the selection of ROIs in pixel by pizel approach.
Did you use PATHOLOGY report for the final analysis in order to define benign versus malignant tissue? Did you exclude from the analysis cases with microcarcinoma, in which the FNA sample could have missed the cancer spot, and to influnec the final accuracy of the method ( to analyse a bening lession considered due to the adiacent microcarcinoma a final cancer case?)
What do you understand under ........ fulle agrees with histopathology/follow up... ( page 2 R 96)
Response 2.3: We did not select a fixed number of ROIs, nor did we perform one ROI per FNA or different ROIs from different FNAs from the same patient. What we did was the annotation by the pathologists of ROIs containing aggregates of thyroid cells (either benign or malignant) from each sample, which depending on the cellularity of the specimen could potentially end in a variable number of final ROIs per sample, as specified in page 7 at the end of the discussion. Moreover, we described this approach in the Methods section, at the end of the paragraph “4.3. MALDI-MSI”, as well as extensively explaining the details of the method in our previous experience (https://www.mdpi.com/2072-6694/11/9/1377/htm).
Regarding your question on the selection of ROIs in the pixel-by-pixel approach, while the analysis “guided” by the ROIs annotated by pathologists was focused on data coming exclusively from the ROIs, in the pixel-by-pixel approach the entire virtual image of the sample was divided in pixels and each of these was analyzed and classified as benign/malignant. This means that for the pixel-by-pixel analysis the detection of ROIs was not used and is not essential for this kind of approach.
Regarding the reference that has been used to assess the benign/malignant nature of the sample, for benign (THY2) cases adequate clinical and ultrasound follow-up (min 24 months) was used to confirm the benign nature of the nodule. For cases with >THY3 final cytology report, thyroidectomy and thus final surgical pathology was used as reference for the final diagnosis of the case. These details are reported in page 8, in Materials and Methods, Patients section. Regarding the “microcarcinoma” point, we analysed only cases with a surgically proven carcinoma diagnosis (independently from the size) and a corresponding cytological pre-surgical sample with at least some cells representative of the lesion.
With “fully agreed with histopathology/follow-up” in the Results, “2.3. MALDI-MSI to classify thyroid nodules: training set” section, we meant that the ROI approach showed the highest concordance with the final “classification” of the case based on histopathology and/or follow-up.
Point 2.4: Training set =
You are stating that 70 FNAs , from 70 different nodules were used for the analysis. If the different ROI.s used to analyse each nodule, offered diveregent results, how was the final decision = majority, worse scenario, minimal thershold. This detail is really under in this section.
Response 2.4: The presence of at least one third (33%) of malignant ROIs within each nodule was used as a threshold to classify the nodule as malignant. Thus, we opted for a criterium based on a minimal threshold of malignant ROIs as a reasonable compromise between the two other options you indicated, i.e. majority and worst scenario. The majority, which means a threshold of at least 50% in malignant ROIs, would result in detecting exclusively clear-cut carcinoma (conservative scenario). On the other hand, the worst scenario, which means at least one malignant ROI, would result in an over-pessimistic situation, in which almost all nodules would be classified as malignant. A sentence has been added in the Result section 2.3, that anticipated the criterium used in the classification of nodules based on ROIs, which is reported in the Statistical Analysis section: “In the ROIs approach, the presence of at least one third of ROIs classified as malignant, within sample, was used as threshold to classify the nodule as malignant.”
Point 2.5: What is the reason for removing the borderline lessions from the final analysis? In the diagnostic, the use of MALDI MSI is usefull in pre preoperative stage, when there is no clue regarding the final result. The validation of the method should be made on all casses, since AF, NIFTP and WDT UMP can be seen under BETHESDA III and IV citological results. Excluding thees ecase in the validation does falsy increase the accuracy of the method.
Response 2.5: Thanks for the observation. Actually, removing cases with “borderline” histology (e.g. NIFTP, FA, WDT UMP) led to increase the specificity of the method in our validation set, not changing the sensitivity. However, we did not remove these cases a priori from the analysis. In page 4, Results, “2.4. MALDI-MSI to predict diagnosis: validation set” section, the model has been tested on a validation cohort of 170 cases, comprising all the samples, from rich and cellular to scant, encompassing also “bordeline” histologies. This led to 83% Sp and 43% Sn. Then, analyzing which major errors were made by the model we noted that some of the common mistakes were made on these “borderline” histologies, and we tried to measure the accuracy of the model excluding these (few) cases. This was not made to “falsely” increase the performance, but instead to stress the actual limits of a model that still requires to be further fed with these borderline cases to be fully ready for clinical implementation. Indeed,
DISCUSSIONS
Point 3.1: In intermediate cytology cases,2 years of follow with is not definitivelly a certitude for benign lessions. When you try to validate a new method/technoque, referral must be made with definitive diagnostic cases.
Response 3.1: We agree with the reviewer that 2 years of clinical and ultrasound follow-up are not comparable to the final histological assessment of the thyroid to confirm the benign nature of the nodule. However, THY2 cases do not generally have indications for surgery, if not for clinical (compression, refractory hyperfunction) reasons. Thus, to consider those nodules as benign international thyroid associations/organizations recommends at least 12 months of follow up (https://pubmed.ncbi.nlm.nih.gov/27167915/). In the proposed work, we doubled the time of observation to further ensure that the benign nature of nodules is confirmed.
Point 3.2: A compoarision bewteen pixel by pixel approach with the evaluation of lipid content ( as the DeHoog research).
Response 3.2: Thanks for your suggestion, we have emphasized the comparison based on a pixel-by-pixel approach between our work and the DeHoog research. We have changed the sentence already mentioned along the text in the Discussion section, as follow: “Our results on FNA needle washes that contain adequate thyroid cells or MALDI-MSI signals are not completely in line with those obtained by DeHoog (2019) (12) who achieved a sensitivity of 96% (95%CI = 79–99%) and a of specificity 91% (95%CI = 77–97%) on 58 FNA smears, focusing on lipid content, using a pixel-by-pixel approach. It is important to highlight that 1) DeHoog used a molecular approach based on metabolic analysis (desorption electrospray ionization mass spectrometry, DESI-MS) on FNA smears; and 2) when considering the precision in the estimates, DeHoog’s and our 95% CIs overlap.”
Point 3.3: I do support that the technique could find an application in reconfirmim benign lessions, even if the diagnostic capacity of PTC versus AF/NIFT P is difficult.
Response 3.3: Thanks for your support. Actually, as we stated along the text, the current performances of the proposed model find a good application in confirming the benign nature of nodules, and further analysis and training of the model on a larger cohort comprising more “borderline” histologies (e.g. FA, NIFTP, WDT-UMP) could further increase the capabilities of MALDI in triaging these nodules.
Reviewer 2 Report
Is the sample size adequate for this study? This Reviewer sees no mention of sample size calculation or of a post-hoc power calculation.
This Reviewer wonders about cost issues - in microbiology analogous techniques are considered to lead to cost savings [J Clin Microbiol. 2015; 53: 2473–2479.] - maybe the authors could add a note.
Author Response
Response to Reviewer 2
Point 1: Is the sample size adequate for this study? This Reviewer sees no mention of sample size calculation or of a post-hoc power calculation.
Response 1: The proposed study was funded by the AIRC (Associazione Italiana per la Ricerca sul Cancro, MFAG 2016 grant, n.133/ 7–2-2017) and the sample size calculation was part of the grant proposal. The study foresaw the enrollment of a total of 240 THY2 (HP) and THY5 (PTC) patients, according to a ratio 2:1 between THY2 and THY5 (160 THY2 and 80 THY5), to reach a 90% power to detect a 1.5-fold change in the mean intensities of the two groups, with an α error of 0.001 and assuming one technical replicate, two-sided test and the variance in the log-peak equal to 0.9. Thus, this was a highly powered study for the discovery of new markers based on proteomic profiling that properly controls for the False Discovery Rate due to multiple testing. The training set we have actually used in the process of model building involved a total of 431 THY2 ROIs (381 HP, 50 HT) and 174 THY5 ROIs (PTC). So, using ROIs instead of patients has allowed us to work on an even larger sample than postulated.
Point 2: This Reviewer wonders about cost issues - in microbiology analogous techniques are considered to lead to cost savings [J Clin Microbiol. 2015; 53: 2473–2479.] - maybe the authors could add a note.
Response 2: Thank you for the input. The following sentence has been add in the discussion section: “Indeed, once the MALDI-MSI approach will became a first-line diagnostic tool for indeterminate cases in the routine workflow of thyroid FNA, an easy-to-use system will be realized, similarly to what has been already implemented in the microbiology units (MALDI-Biotyper (21)). The system will thus increase diagnostic accuracy, while reducing the costs related to unnecessary surgery and will improve the cost-effectiveness of post-treatment follow-up examinations.”